# Targeting earlier diagnosis: What symptoms come first in Degenerative Cervical Myelopathy?

Colin F. Munro[1], Ratko Yurac[2,3], Zipser Carl Moritz[4], Michael G. Fehlings[5,6,7], Ricardo Rodrigues-Pinto[8,9], James Milligan[10], Konstantinos Margetis[11,12], Mark R. N. Kotter[1,13], Benjamin M. Davies[1,13,14]*

1 Division of Neurosurgery, University of Cambridge Department of Clinical Neurosciences, Cambridge, Cambridgeshire, United Kingdom, 2 Department of Traumatology, Spine Unit, Clinica Alemana de Santiago SA, Vitacura, Santiago, Chile, 3 Department of Orthopedic and Traumatology, Desarrollo University Faculty of Medicine, Las Condes, Chile, 4 University Spine Center, Balgrist University Hospital, Zurich, Switzerland, 5 Division of Neurosurgery, Department of Surgery, University of Toronto, Toronto, Ontario, Canada, 6 Institute of Medical Science, University of Toronto, Toronto, Ontario, Canada, 7 Division of Neurosurgery, Toronto Western Hospital Krembil Neuroscience Centre, Toronto, Ontario, Canada, 8 Department of Orthopaedics, Spinal Unit (UVM), Centro Hospitalar Universitário do Porto EPE, Porto, Portugal, 9 Universidade do Porto Instituto de Ciencias Biomedicas Abel Salazar, Porto, Portugal, 10 McMaster University Department of Family Medicine, Hamilton, Ontario, Canada, 11 Department of Neurosurgery, Mount Sinai Hospital, New York, New York, United States of America, 12 Department of Neurosurgery, Icahn School of Medicine at Mount Sinai, New York, New York, United States of America, 13 Myelopathy.org, Charity for Degenerative Cervical Myelopathy, Cambridge, Cambridgeshire, United Kingdom, 14 AOSpine International, RECODE DCM Incubator, Diagnostic Criteria, Davos, Graubünden, Switzerland

* bd375@cam.ac.uk

**Data Availability Statement:** Data cannot be shared publicly because it contains potentially identifying and sensitive patient information. Data

## Abstract

### Background

Degenerative cervical myelopathy (DCM) is a common and disabling condition. Early effective treatment is limited by late diagnosis. Conventional descriptions of DCM focus on motor and sensory limb disability, however, recent work suggests the true impact is much broader. This study aimed to characterise the symptomatic presentation of DCM from the perspective of people with DCM and determine whether any of the reported symptoms, or groups of symptoms, were associated with early diagnosis.

### Methods

An internet survey was developed, using an established list of patient-reported effects. Participants (N = 171) were recruited from an online community of people with DCM. Respondents selected their current symptoms and primary presenting symptom. The relationship of symptoms and their relationship to time to diagnosis were explored. This included symptoms not commonly measured today, termed 'non-conventional' symptoms.

### Results

All listed symptoms were experienced by >10% of respondents, with poor balance being the most commonly reported (84.2%). Non-conventional symptoms accounted for 39.7% of symptomatic burden. 55.4% of the symptoms were reported as an initial symptom, with

is available from the Myelopathy.org charity (contact via info@myelopathy.org) for researchers who meet the criteria for access to confidential data.

**Funding:** MRNK is supported by the National Institute for Health Research (NIHR) Brain Injury MedTech Co-operative based at Cambridge University Hospitals NHS Foundation Trust and BMD via a NIHR Clinical Doctoral Research Fellowship. The views expressed in this publication are those of the authors and not necessarily those of the NHS, the National Institute for Health Research or the Department of Health and Social Care.

**Competing interests:** I have read the journal's policy and the authors of this manuscript have the following competing interests: CFM has declared that no competing interests exist. RY has declared that no competing interests exist. ZCM has declared that no competing interests exist. MGF currently serves as an academic editor at PLOS ONE. RRP has declared that no competing interests exist. JM has declared that no competing interests exist. KM has declared that no competing interests exist. MRNK has declared that no competing interests exist. BMD is supported by NIHR POLYFIX DCM and NIHR Clinical Doctoral Research Fellowship grants. BMD is a founder of MoveMed (a digital therapeutics platform which develops assessments and treatments using software). The funders had no role in study design, data collection and analysis, decision to publish, or preparation of the manuscript. This does not alter our adherence to PLOS ONE policies on sharing data and materials.

neck pain the most common (13.5%). Non-conventional symptoms accounted for 11.1% of initial symptoms. 79.5% of the respondents were diagnosed late (>6 months). Heavy legs was the only initial symptom associated with early diagnosis.

## Conclusions

A comprehensive description of the self-reported effects of DCM has been established, including the prevalence of symptoms at disease presentation. The experience of DCM is broader than suggested by conventional descriptions and further exploration of non-conventional symptoms may support earlier diagnosis.

## Introduction

Degenerative Cervical Myelopathy (DCM) is a progressive condition that occurs when the cervical spinal cord is compressed by degenerative changes in surrounding structures [1].

DCM treatment is largely restricted to surgery that aims to alleviate compression of the spinal cord. Recent large prospective studies have demonstrated surgery is able to stop disease progression and offer a meaningful, albeit incomplete recovery [2, 3]. The amount of recovery is hypothesized to be dependent on the severity of existing damage. Consequently, time to treatment has emerged as an important predictor of treatment response [4], with the latest analysis indicating a preoperative duration of symptoms less than 4 months offers the most favourable outcome [5].

Whilst not applicable to the full spectrum of DCM, with mild forms of the disease amenable to a watch and wait approach [6], for those requiring treatment any such prompt intervention target is currently undeliverable. This is due to long delays in diagnosis, on average 2–5 years [7, 8]. This is contributing to the significant residual disability in DCM, with dependence and unemployment prevalent [9] and enabling timely treatment underpins many of the top research priorities identified by AO Spine RECODE-DCM [10].

Consequently, improving time to diagnosis is an attractive target for improving outcomes immediately in DCM. However, the factors driving missed and delayed diagnosis are poorly characterised at present and difficult to investigate. Whilst a poor awareness and understanding of the disease are undoubtedly factors [1, 11], the information required to support early diagnosis, including knowledge of the early symptoms, or the key differentials or at risk populations, is yet to be established.

Conventional descriptions of DCM have focused on motor and sensory disability to the limbs [12, 13]. This is reflected in the assessment of DCM for clinical research [14] and clinical care [15]. However, we have recently developed a long-list of patient reported symptoms in DCM [16], which broadens the potential impact of DCM. This is also a unique dataset as outcomes are based on the patient's own wording.

This study aimed to characterise the symptomatic presentation of DCM from the perspective of people with cervical myelopathy [17] and determine whether any of the reported symptoms, or groups of symptoms, were associated with early diagnosis.

## Methods

This study aimed to determine which of the patient reported symptoms in DCM were associated with earlier diagnosis. The study used a long-list of patient reported outcomes, previously established [16]. This study was conducted with ethical approval from the University of

Cambridge. It is reported in accordance with the recommendations for Conducting and REporting DElphi Studies (CREDES) [18] and Checklist for Reporting Results of Internet E-Surveys (CHERRIES) [19].

## Survey design

An internet survey was developed using SurveyMonkey (California, USA), using a comprehensive list of patient reported effects of DCM, previously established [16, 20]. In brief, this process had used semi-structured interviews with people with DCM (N = 5) and carers (N = 3) to identify effects of DCM [20]. These effects were then presented to a separate and larger cohort of sufferers via an initial internet survey (N = 224) (S1 Appendix). The survey was advertised via Myelopathy.org, an international charity for people with DCM. Respondents were asked to confirm whether they suffered from 36 shortlisted outcomes, but also given the opportunity to submit additional suggestions (S1 Table). These were then processed by investigators to produce a 'longlist' of 56 patient outcomes (S1 Table) [16]. Outcomes were generated based directly on sufferers' wording and goes beyond the common and existing descriptions of DCM, which are the focus of current myelopathy assessment [14].

The final survey was formed of three sections. Initially participants were provided with an overview of the study and definition of DCM. By continuing into the survey, participants were confirming their diagnosis of DCM and providing informed consent to participate. Respondents were asked a series of sampling questions, including age, gender, length of time between symptom onset and diagnosis, length of time lived with DCM, history of surgical treatment and disease severity as measured using the p-mJOA (patient derived modified Japanese Orthopaedic Association) score. The mJOA is the international standard for assessment of disease severity [6, 13], and the p-mJOA a validated patient reported version [21]. Finally, the list of 56 patient reported outcomes were incorporated into a matrix, with respondents asked: "What symptoms do you currently experience due to DCM?" and "Which of these was the first symptom you experienced as a result of DCM?". For this latter question, only one symptom could be selected. If they experienced more than one symptom initially and could not remember which came first, they were asked to select the most significant one at the time. The survey consisted of 16 questions over 10 pages (S2 Appendix).

Participation was voluntary and advertised using Myelopathy.org, an international nonprofit organisation dedicated to promoting understanding and awareness of DCM, to help people with DCM, professionals and supporters. Participants from the previous (first-round) internet survey were invited by email to participate in this follow-up survey. Surveys could not be edited once they had been completed. No incentives were offered for completion of the surveys.

IP addresses were screened to identify potential duplicate responses. If multiple entries from the same IP address were discovered, only one response was included, unless the responses gave different email addresses or had significantly different demographic data (age and sex), in which case it was assumed to be a separate individual on the same device. The more completed response of the duplicates was included, if both were equally complete, the response with the earliest end date and time was included.

## Analysis

JASP software (Version 0.13.1) was used for statistical analysis. The Shapiro-Wilks test was used to assess for parametric distribution of numerical data sets. The Mann-Whitney U test was then used to compare the means of two non-parametric distributions whilst a Two Tailed T-test used to compare the means of two parametric distributions. Analysis of variance

(ANOVA) was used to compare the means of 3 or more parametric distributions, whilst the Kruskal-Wallis test was used to compare the means of 3 or more non-parametric distributions. Chi-Square test for association was used to assess the relation between categorical variables. A result was taken as significant when $p < 0.05$.

Odds ratios, and their corresponding 95% confidence intervals, were used to assess the association between the presence of an initial symptom and early or late diagnosis, with a confidence interval not encompassing 1 taken as a significant result.

Only complete surveys were used for data analysis. Surveys were categorised as incomplete (no questions answered after respondent confirmed they had DCM), partially complete (respondent only answered demographics questions) or fully complete (respondent answered all questions). The demographic characteristics of partially complete and fully complete responses were compared in order to evaluate the potential impact of missing data. Demographic characteristics were also compared to the previous round.

In order to assess the representation of DCM amongst the recruited cohort, current symptoms were matched to those recorded in the AO Spine prospective observational study (N = 679) of sufferers undergoing surgery, and their prevalence compared (S2 Table and S1 Fig).

Early diagnosis was defined as <6 months from the onset of symptoms, and late diagnosis >6 months. This is based upon analysis showing treatment within 6 months is associated with improved outcomes [4, 22].

To evaluate the impact of symptoms less commonly acknowledged professionally with DCM, patient reported outcomes were divided into either 'conventional' or 'non-conventional' symptoms based on their acknowledgement or not within DCM review articles [12]. To evaluate the impact of key areas of effect, conventional symptoms were further subdivided into motor, sensory, pain and autonomic categories. The motor and sensory categories were then subdivided anatomically into upper and lower limb groups. This was based on the structure of the m/JOA assessment, the current gold-standard assessment for DCM [6, 13]. The non-conventional group was sub-divided into: sensation/pain, movement disorder, gastro-intestinal, respiratory, cranial and psychosocial. This grouping was agreed by the authors, based on common systems. The final classification is split between Tables 1 and 2, showing conventional and non-conventional symptoms respectively.

## Results

Overall, 189 unique individuals accessed this survey. This included 78 respondents who had participated in the previous project (Fig 1).

**Table 1. Survey symptom classification–conventional symptoms.**

| Conventional | | | | | |
| --- | --- | --- | --- | --- | --- |
| Motor | | Sensory | | Pain | Autonomic |
| LL Motor | UL Motor | UL Sensory | LL Sensory | | |
| Poor balance* | Clumsiness* | Pins and needles in your hand* | Leg numbness* | Neck pain* | Difficulty emptying bladder* |
| Lack of control of legs* | Reduced dexterity* | Lhermitte's phenomena* | Pins and needles in your leg* | Shoulder pain* | Faecal incontinence |
| Leg stiffness* | Reduced grip strength* | Hand numbness* | | Neck stiffness* | Urinary incontinence |
| Heavy legs* | Arm stiffness* | Arm numbness* | | Neck clicking* | Waking to go to the toilet |
| Dragging legs* | | Pins and needles in your arm* | | Back pain* | Erectile Dysfunction |
| Falls* | | | | Arm pain* | |
| | | | | Allodynia | |
| | | | | Leg pain | |

*Symptoms that were reported as initial symptoms.

**Table 2. Survey symptom classification–non-conventional symptoms.**

| Non-Conventional | | | | | |
| --- | --- | --- | --- | --- | --- |
| **Sensation / Pain** | **Movement Disorder** | **Gastro-Intestinal** | **Respiratory** | **Cranial** | **Psychosocial** |
| Altered temperature sensation* | Leg shaking* | Choking/swallowing problems* | Exertional breathlessness | Dizziness* | Symptom variability day by day |
| Female sexual dysfunction | Muscle spasms or twitches (arms)* | Constipation | Difficulty breathing when lying flat | Headache* | Impaired cognition |
| Hot flushes and/or sweating | Hand shaking* | Nausea and vomiting | | Tinnitus | Anxiety |
| Abdominal pain | Muscle spasms or twitches (legs) | | | Eyesight problems | Fatigue |
| Face pain | | | | Face numbness | Insomnia |
| | | | | | Symptom variability hour by hour |
| | | | | | Depression/low mood |

*Symptoms that were reported as initial symptoms.

Demographic information was not significantly different between complete and partially complete responses. Overall, survey respondents matched those of the previous round, with the exception of surgical history: Respondents of round 3 were more likely to have had surgery for their DCM (p = 0.04) (Table 3).

## Current symptoms

All symptoms were currently experienced by at least 10% of respondents, with each respondent on average experiencing 27 different symptoms. The most commonly reported individual symptoms were poor balance (84.2%), clumsiness (80.7%), neck stiffness (78.4%), reduced grip strength (76.6%), and hand numbness (75.4%) (Fig 2). Current symptom burden closely matched that experienced by sufferers participating in the AO Spine prospective observational study (S1 Fig).

The proportion of all reported symptoms attributable to each symptom group is shown in Fig 3. Conventional symptoms account for 60.3% of symptomatic burden and non-conventional symptoms account for 39.7%.

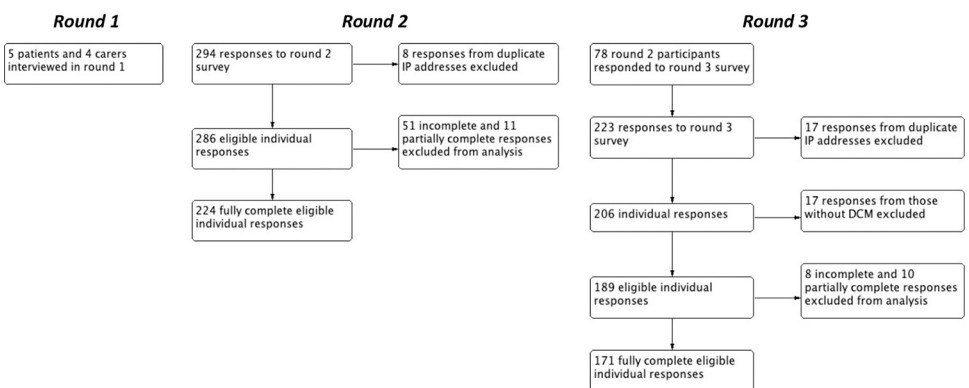

**Fig 1. Flowchart of the study design.** The findings from previous semi-structured (Round 1) interviews [20] and paired internet survey (Round 2) [16], were used to produce this survey (Round 3).

**Table 3. Comparison of demographics of survey respondents in round 2 and round 3.**

| Respondent Demographics: | Round 2 (N = 224): | Round 3 (N = 171): | P value |
|---|---|---|---|
| Mean age (years) | 56.6 | 53.9 | 0.63 |
| Female/Male (%) | 75.9/24.1 | 73.9/26.3 | 0.62 |
| Surgery/No Surgery (%) | 62.1/38.0 | 71.9/28.1 | 0.04* |
| Mean time to diagnosis (years) | 4.9 | 3.9 | 0.23 |
| Early/Late Diagnosis (%) | 21.0/79.0 | 20.5/79.5 | 0.90 |
| Mean length of time with DCM (years) | 8.2 | 6.8 | 0.43 |
| Mean total mJOA | 11.6 | 11.5 | 0.93 |
| Mean upper limb motor mJOA | 3.6 | 3.6 | 0.99 |
| Mean lower limb motor mJOA | 4.3 | 4.2 | 0.67 |
| Mean upper limb sensory mJOA | 1.7 | 1.7 | 0.50 |
| Mean sphincter dysfunction mJOA | 2.1 | 2.1 | 0.93 |

Except for surgical history, there were no significant difference between cohorts.

*$<$0.05

**Fig 2. Bar chart of the prevalence of symptoms.** The number of respondents reporting each symptom are displayed at the end of the bar.

## Proportion of overall symptomatic burden attributable to each symptom category

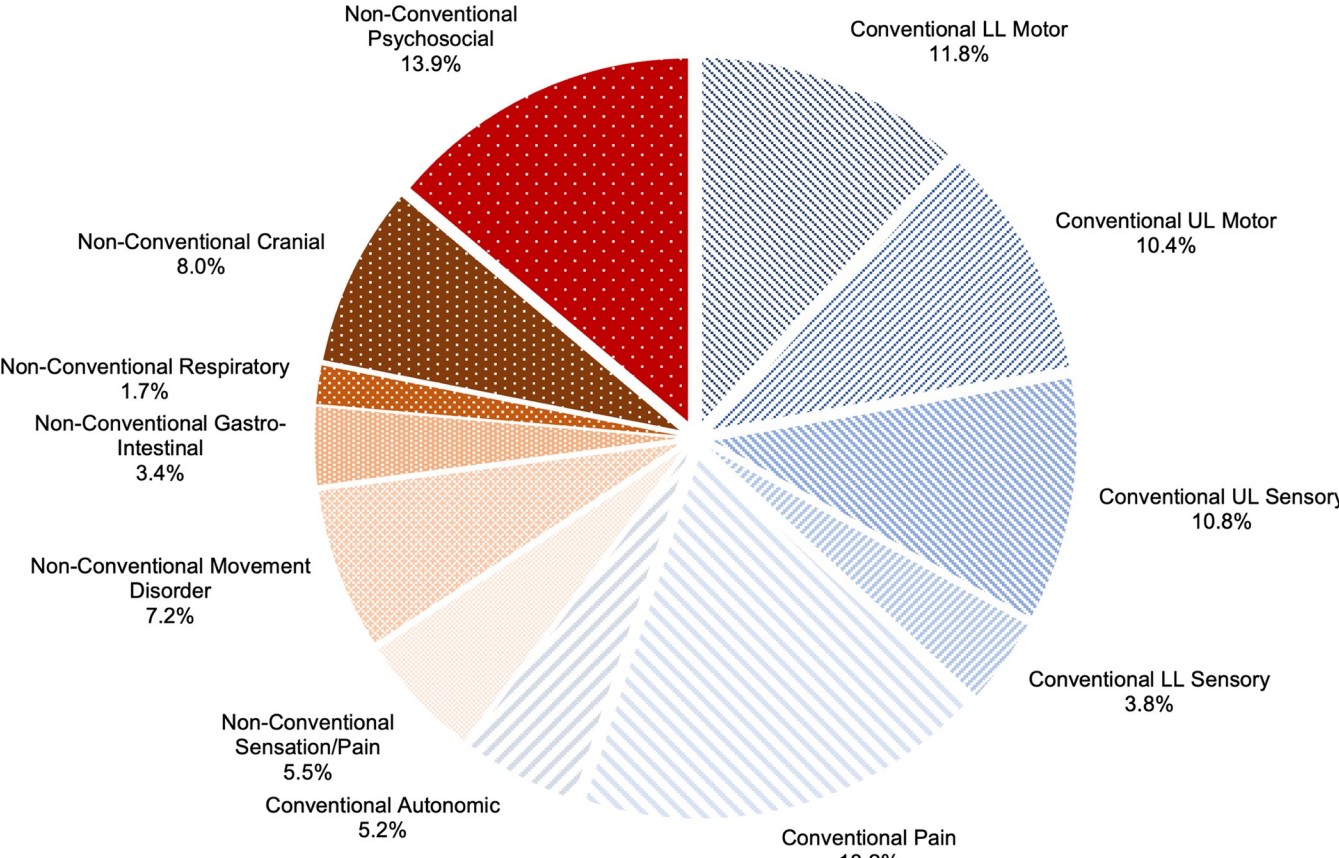

**Fig 3. Pie chart illustrating the proportion of overall symptoms reported that were attributable to each symptom category.** Conventional symptom segments are illustrated with lines and non-conventional symptom segments with dots.

### Initial symptoms

Of the 56 listed symptoms, 31 (55.4%) were reported as an initial symptom. The most commonly reported individual initial symptoms were: neck pain (13.5%), shoulder pain (8.8%), pins and needles in the hand (7.0%), arm pain (5.3%) and Lhermitte's phenomena (5.3%) (Fig 4). The percentage of respondents with an initial symptom from each symptom category is shown in Fig 5. Conventional symptoms account for 88.9% of initial symptoms and non-conventional symptoms account for 11.1%.

### Association between initial symptoms and time to diagnosis

The mean time to diagnosis was 46.4 months, with 20.5% (35/171) diagnosed early (within 6 months) and 79.5% (136/171) diagnosed late (after 6 months).

Fig 6 shows the association between individual, initial symptoms and the likelihood of an early or late diagnosis. Heavy legs was the only symptom significantly associated with early diagnosis (95% Confidence Interval <1). No initial symptoms were significantly associated with late diagnosis.

Initial symptoms were categorised into the predefined symptom domains. Fig 7 shows their association with early or late diagnosis. No initial symptom group significantly favoured an

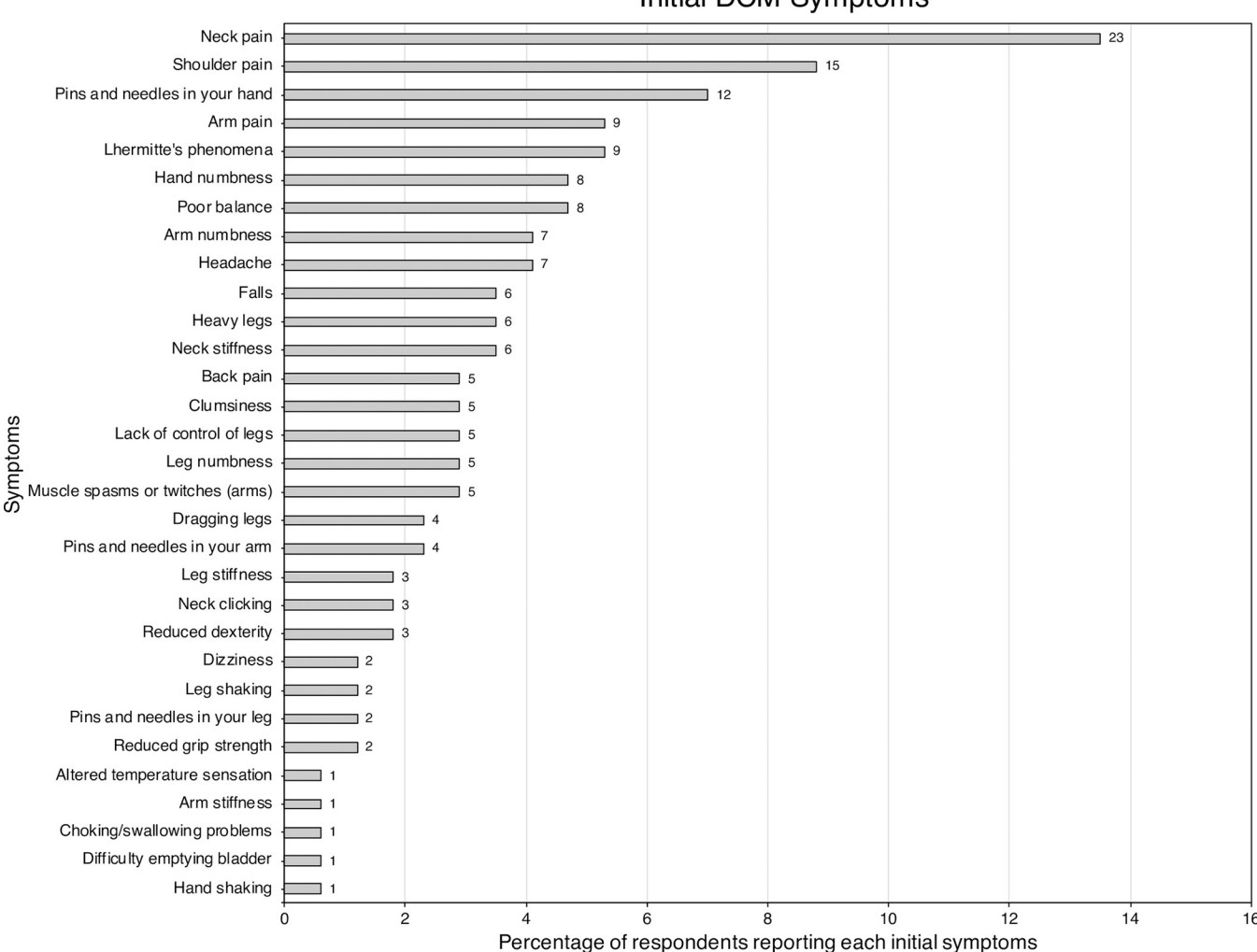

**Fig 4. Bar chart of the prevalence of initial symptoms.** The number of respondents reporting each initial symptom are displayed at the end of the bar.

early or late diagnosis, although there was a trend for non-conventional cranial symptoms to be associated with a late diagnosis.

## Discussion

This is the first study to explore the patient reported experience of DCM at presentation. The findings of this survey indicate that patients experience a far greater breadth of symptoms than are commonly considered in textbooks [12], or evaluated in clinical research [14, 22] or clinical care [15]. Whilst this study indicates that a sub-selection may be particularly relevant for detection, they remain diverse and non-specific. Only 'heavy legs' appeared specific for early diagnosis.

### Limitations

The findings of this study must be considered in the context of its methodology. As an open, internet recruited survey of self-selected people with DCM, reporting their retrospective

## Proportion of initial symptoms attributable to each symptom category

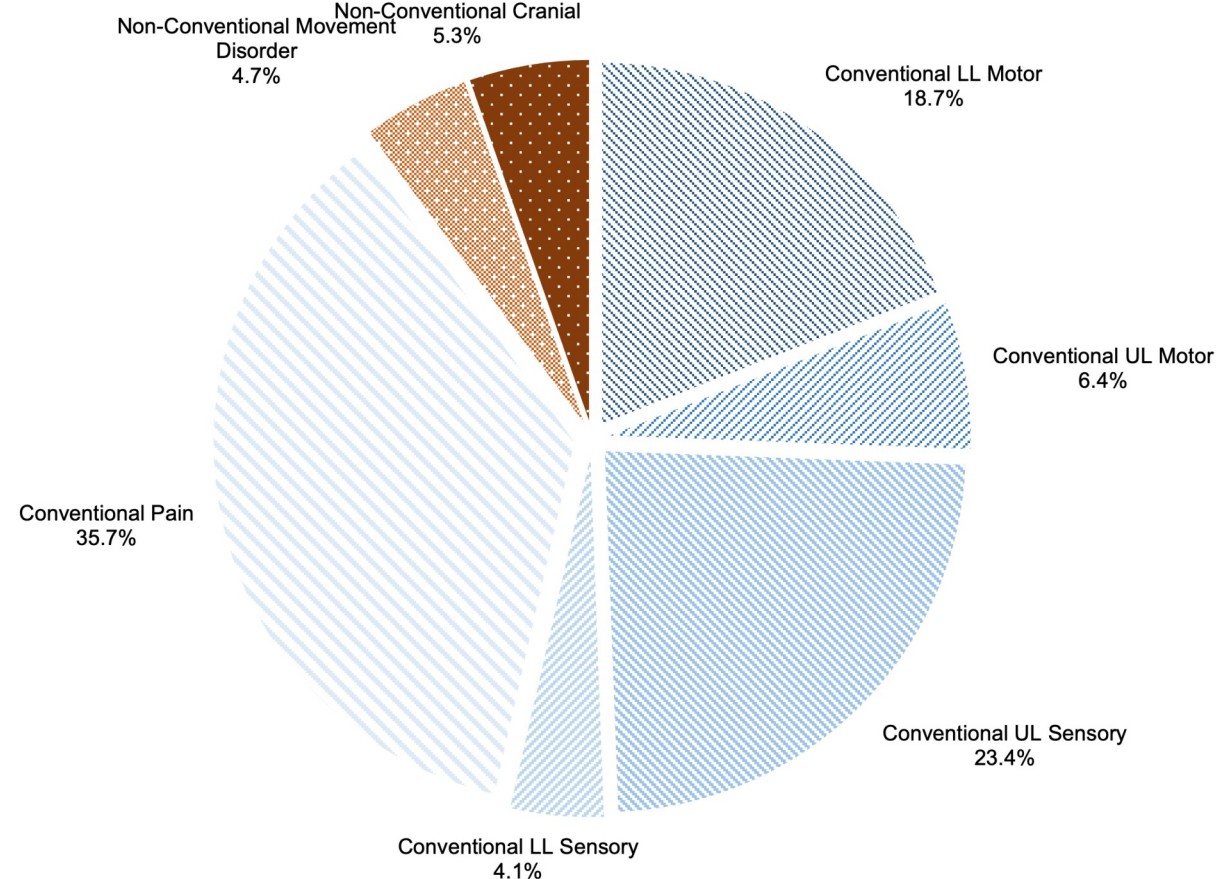

**Fig 5. Pie chart illustrating the proportion of initial symptoms that were attributable to each symptom category.** Conventional symptom segments are illustrated with lines and non-conventional symptom segments with dots. Categories with N ≤ 1 have been left out for the purposes of clear illustration.

experience, the results are at risk of sampling and recall bias. That said, a number of design factors and findings are reassuring that this is unlikely to be significant.

Firstly, internet surveys are increasingly recognised as an effective means of reaching representative samples of a disease [23–26]. For example, they are the mainstay of core-outcome set initiatives to define symptom burden and a recent exercise in inflammatory bowel disease, using cross-validation with an individual's health records, found self-reporting is accurate [27]. Further, in our survey participants were presented with a description of DCM, and asked to confirm their diagnosis, making it unlikely people without a diagnosis of DCM participated.

Secondly, the disease characteristics of participants, including symptom profiles, matched those identified within the high-quality AO Spine observational studies [5] (S1 Fig). Further, whilst respondents were more likely to be female, as has been the case with previous surveys using Myelopathy.org, a DCM charity [9, 16], the cohort demographics also matched the clinical series [5] (S2 Table). This inconsistency is thought to reflect the increased participation of women in online health-communities [28].

Traditionally a DELPHI process will recruit respondents only once, inviting and measuring dropout for each consecutive round. As an additional adjunct, the third-round survey was an

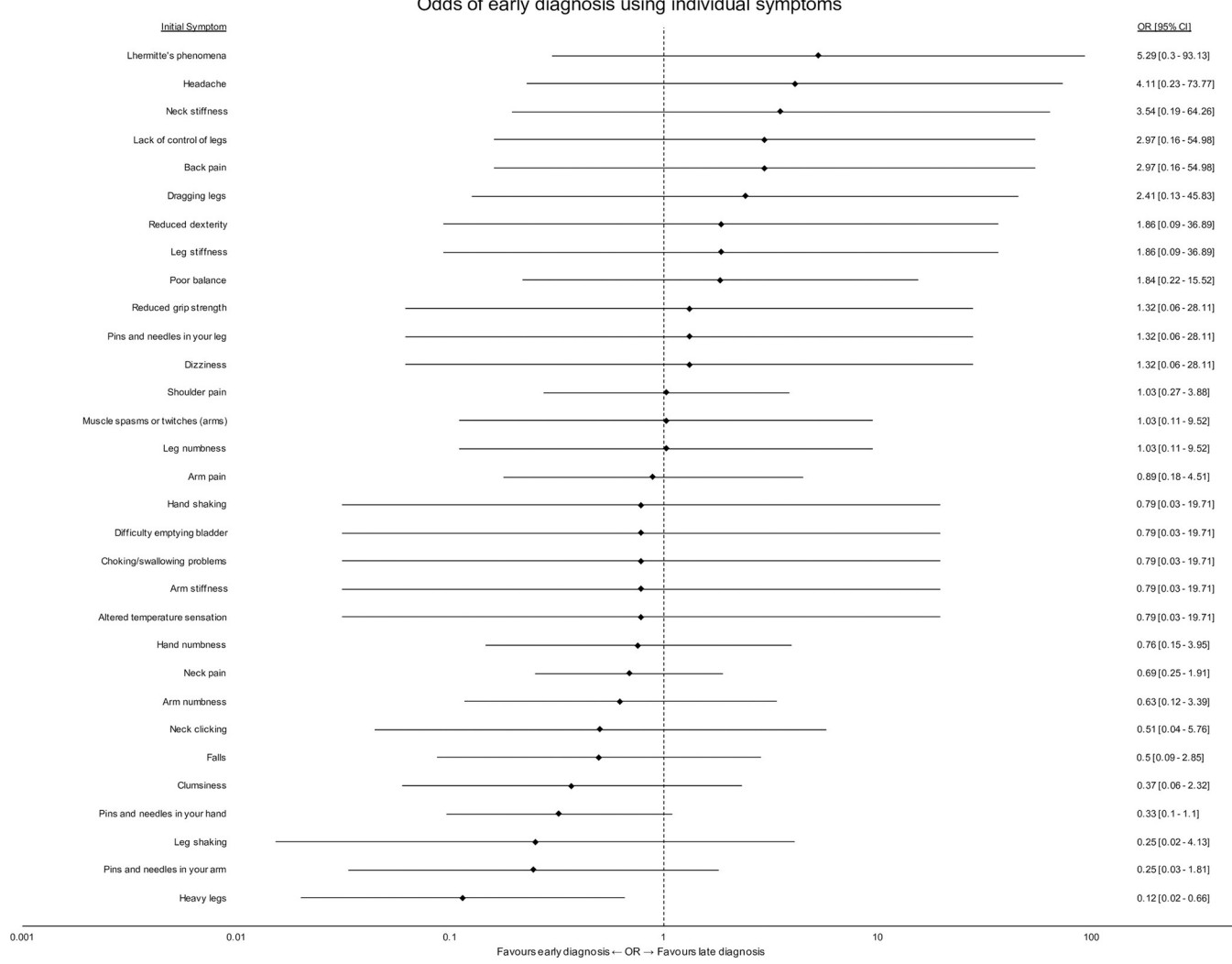

**Fig 6. Odds of early diagnosis using individual symptoms.** A forest plot of individual odds ratios for each initial symptom. Error bars represent 95% confidence intervals of the odds ratio.

open survey to improve the response rate. However, the consistent sampling demographics provide some reassurance that this is unlikely to have influenced the findings.

Due to the very varied nature of DCM presentation, the absolute numbers of respondents presenting with any one symptom at presentation was often small. This likely resulted in the study being underpowered to find significant associations between initial symptoms and early or late diagnosis. It was therefore also felt unsuitable to explore with modelling. Thus, one needs to be careful to avoid a Type II error in concluding that, with the exception of heavy legs, no initial symptoms are associated with either early or late diagnosis.

## Interpretation

**Pain and upper limb sensory symptoms predominate initially.**   Whilst there was substantial variability in the individual presenting symptoms, the majority (59.1%) of respondents first experienced a symptom which could be classified as a conventional pain or upper limb sensory symptom, indicating a potential focus point. Behrbalk et al (2013) in their description

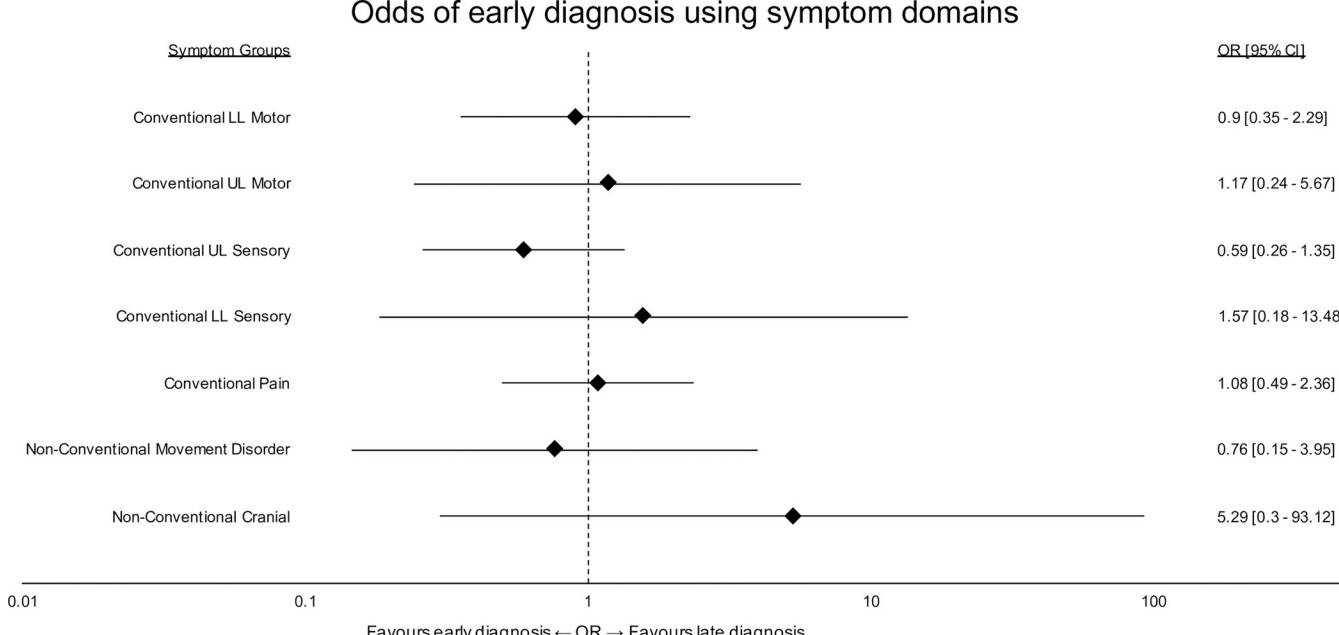

**Fig 7. Odds of early diagnosis using symptom categories.** A forest plot of individual odds ratios for each symptom category. Error bars represent 95% confidence intervals of the odds ratio. Categories with N ≤ 1 have been left out for the purposes of clear illustration.

of diagnostic delay, found 43% of patients were initially diagnosed and sometimes treated for carpal tunnel syndrome [7]. Nevertheless, the diagnostic utility of these symptoms is unclear, given the one year incidence for neck pain, in the general population, ranges from 10 to 21% [29] and that general practitioners are typically consulted 7 times a week for neck or upper extremity complaints of various causes [30]. Whilst pain is a common prompt to seek health-care assessment in general [31, 32], its experience here was not associated with earlier diagnosis in this series. Consequently, the absence of these symptoms may instead be useful for ruling out DCM, given their high sensitivity, but low specificity, for the condition [33, 34].

**Are lower limb symptoms helpful for diagnosis?** Gait dysfunction is considered one of the earliest clinical manifestations of DCM [35]. It was the most common (60%) first symptom of myelopathy in a prospective observational study of asymptomatic spinal cord compression [36], and in their review of the diagnostic accuracy of DCM symptoms, Mizer et al [34] found that difficulty in walking for 15 minutes was one of 4 symptoms with a positive likelihood ratio for DCM greater than 5.

Gait dysfunction was not individually matched in this survey. The 'conventional lower limb motor symptoms' group would contribute to gait dysfunction and made up 18.7% of initial symptoms. Of the individual symptoms, only 'heavy legs' was associated with early diagnosis. This is a similar finding to that of Hilton et al. who found that subjective imbalance was the only symptom associated with a shorter referral time between primary and secondary assessment [15]. Whilst this could be confirmation of its importance to detection, it could also represent a selection bias by professionals: driven by the socio-economic consequence of falls [37], their investigation has well-defined pathways from primary to secondary care [15]. This contrasts with the lack of a clear or unified referral pathway for most DCM symptoms, with patients being seen by neurologists, orthopaedics, pain specialists, rheumatologists and geriatricians, contributing to delays in assessment and treatment [8].

**Non-conventional symptoms are common and overlooked. Could they have a role in early detection?**    In this study, symptoms which were not cited in narrative review articles on DCM, were categorised as "non-conventional". This included a number of controversial symptoms, including headache, vision and hearing impairments and dizziness. A number of hypotheses are proposed, including altered signalling via the sympathetic chain (the so called Barré-Liéou "syndrome") or facial symptoms via involvement of the spinal nucleus of the trigeminal nerve [38, 39]. However, these remain hypotheses, as whilst there are numerous descriptions of their association, particularly in the context of cervical spondylosis, the evidence base linking the two remains of low quality [40–43] and as standard DCM assessments do not capture these symptoms, high quality series cannot comment. However, they do demonstrate the high prevalence of co-morbidities in these patient groups, and it is possible these experiences are secondary to different disease processes [5].

Nevertheless, whilst using similar sampling techniques, this is the second cohort in which we have described prevalent non-conventional symptoms [16]. Specifically, in this study 100% of respondents reported at least one of the 26 listed "non-conventional" symptoms and 39.7% of overall symptomatic burden was attributable to non-conventional symptoms. 11.1% of patients reported non-conventional symptoms as the first manifestation of their disease, with the vast majority (89.5%) of these being cranial or movement disorder symptoms, and whilst not shown to have statistical significance in this study, presenting with cranial symptoms was associated with a higher probability of late diagnosis.

Although controversial, this prevalence could have significant value for the detection, and subsequent earlier diagnosis, of DCM, as these symptoms will not occur with many differentials, for example carpal tunnel syndrome. We note, whilst not as comprehensive in their development of associated symptoms, a screening questionnaire based solely on symptoms for the detection of DCM (sensitivity 93.5%; specificity 67.3%) found that the odds ratio of chest tightness, a non-conventional symptom not identified by this study, in myelopathy patients compared with controls was 22.9 [44]. Supporting the proposition that non-conventional symptoms may have a role to play in the earlier diagnosis of DCM.

## Conclusions

This study has re-confirmed that patients describe a varied experience of DCM, much broader than conventional descriptions, which is also the case from the outset; Non-conventional symptoms comprised 40% of a patient's symptom burden and were experienced by all individuals. Early symptoms most commonly relate to pain or upper limb sensation, although individually heavy legs were the only single symptom associated with early diagnosis.

Understanding how these symptoms can be used to distinguish and diagnose DCM early will require further research, including into their sensitivity and specificity individually but also in combination. This is an active goal of the RECODE-DCM Diagnostic Criteria Incubator, an international working group hosted by Myelopathy.org. Parties interested in supporting this consortia are welcomed.

## Supporting information

**S1 Appendix. Copy of round 2 internet survey.**
(DOCX)

**S2 Appendix. Copy of round 3 internet survey.**
(DOCX)

**S1 Table. Short and longlisted survey outcomes.** *Outcomes added after respondent suggestion in round 2. 2 shortlisted symptoms (italicised) were expanded "anatomically" for the longlist.
(DOCX)

**S2 Table. Comparison of demographics between survey cohort and AO Spine observational study.** Aside from gender and time to diagnosis, the cohorts are closely matched.
(DOCX)

**S1 Fig. Symptom frequency comparison with Tetreault et al. 2018.** Comparison of matched symptoms between this study (white) and the AO Spine prospective observational study of sufferers undergoing surgical treatment for DCM (grey). Error bars represent 95% confidence intervals.
(DOCX)

# Author Contributions

**Conceptualization:** Mark R. N. Kotter, Benjamin M. Davies.

**Data curation:** Colin F. Munro.

**Formal analysis:** Colin F. Munro.

**Funding acquisition:** Mark R. N. Kotter, Benjamin M. Davies.

**Methodology:** Benjamin M. Davies.

**Project administration:** Benjamin M. Davies.

**Resources:** Mark R. N. Kotter, Benjamin M. Davies.

**Supervision:** Mark R. N. Kotter, Benjamin M. Davies.

**Writing – original draft:** Colin F. Munro, Benjamin M. Davies.

**Writing – review & editing:** Colin F. Munro, Ratko Yurac, Zipser Carl Moritz, Michael G. Fehlings, Ricardo Rodrigues-Pinto, James Milligan, Konstantinos Margetis, Mark R. N. Kotter, Benjamin M. Davies.

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
