## [Decision Letter · Decision Letter 0]

28 Oct 2022

PONE-D-22-10626Targeting earlier diagnosis: what symptoms come first in Degenerative Cervical Myelopathy?PLOS ONE

Dear Dr. Davies,

Thank you for submitting your manuscript to PLOS ONE. After careful consideration, we feel that it has merit but does not fully meet PLOS ONE’s publication criteria as it currently stands. Therefore, we invite you to submit a revised version of the manuscript that addresses the points raised during the review process.

Please address the points rised by the reviewer.

We look forward to receiving your revised manuscript.

Kind regards,

Andrea Martinuzzi

Academic Editor

PLOS ONE

Journal Requirements:

“Research in MRNK's laboratory is supported by a core support grant from the Wellcome Trust and Medical Research Council to the Wellcome Trust-Medical Research Council Cambridge Stem Cell Institute.

This report is independent research arising from a National Institute for Health Research Clinician Scientist Award to MRNK (CS-2015-15-023).

4. Thank you for providing the following Funding Statement: 

” BMD is supported by NIHR POLYFIX DCM and NIHR Clinical Doctoral Research Fellowship grants. BMD is a founder of MoveMed (a digital therapeutics platform which develops assessments and treatments using software). The funders had no role in study design, data collection and analysis, decision to publish, or preparation of the manuscript.”

We note that one or more of the authors is affiliated with the funding organization, indicating the funder may have had some role in the design, data collection, analysis or preparation of your manuscript for publication; in other words, the funder played an indirect role through the participation of the co-authors.

If the funding organization did not play a role in the study design, data collection and analysis, decision to publish, or preparation of the manuscript and only provided financial support in the form of authors' salaries and/or research materials, please review your statements relating to the author contributions, and ensure you have specifically and accurately indicated the role(s) that these authors had in your study in the Author Contributions section of the online submission form. Please make any necessary amendments directly within this section of the online submission form.  Please also update your Funding Statement to include the following statement: “The funder provided support in the form of salaries for authors [insert relevant initials], but did not have any additional role in the study design, data collection and analysis, decision to publish, or preparation of the manuscript. The specific roles of these authors are articulated in the ‘author contributions’ section.”

If the funding organization did have an additional role, please state and explain that role within your Funding Statement.

Please also provide an updated Competing Interests Statement declaring this commercial affiliation along with any other relevant declarations relating to employment, consultancy, patents, products in development, or marketed products, etc. 

Reviewers' comments:

Reviewer's Responses to Questions

**Comments to the Author**

1. Is the manuscript technically sound, and do the data support the conclusions?

Reviewer #1: Yes

Reviewer #2: Yes

2. Has the statistical analysis been performed appropriately and rigorously? 

Reviewer #1: Yes

Reviewer #2: Yes

3. Have the authors made all data underlying the findings in their manuscript fully available?

Reviewer #1: Yes

Reviewer #2: Yes

4. Is the manuscript presented in an intelligible fashion and written in standard English?

Reviewer #1: Yes

Reviewer #2: Yes

5. Review Comments to the Author

Reviewer #1: Its very well written article with sound methodology and analysis of the results. It gives good insight into DCM. This helps inn early diagnosis of the DCM. How much of importance to be given to non conventional symptoms?

Reviewer #2: The study design and analysis are well done. I have little to say about this.

The authors focus on the symptoms, and how they do or do not relate to early diagnosis, which is an important finding.

However, the frequency of first symptoms add to the body of knowledge, the top 5 to 6 symptoms that the patient reports, should lead to more in depth assessment of the patient and that general comment or statement is not made in the paper.

For the group writing this paper simply reporting on results should not be the end of the paper, a recommendation or two should come of it.

I would like to see something related to, "if we see …… symptoms, further assessment should be done to confirm diagnosis." An emphasis should be placed on, when upper limb symptoms occur, the neck should be cleared and red flags should be investigated.

I would like to see a little more thought and clinical relevance emphasized in the discussion and conclusion.

6. PLOS authors have the option to publish the peer review history of their article (what does this mean?). If published, this will include your full peer review and any attached files.

Reviewer #1: No

Reviewer #2: **Yes: **Sukhvinder Kalsi-Ryan

---

## [Author Response · Author response to Decision Letter 0]

8 Dec 2022

Editor’s comments:

1. “Please ensure that your manuscript meets PLOS ONE's style requirements, including those for file naming. The PLOS ONE style templates can be found at https://journals.plos.org/plosone/s/file?id=wjVg/PLOSOne_formatting_sample_main_body.pdf and https://journals.plos.org/plosone/s/file?id=ba62/PLOSOne_formatting_sample_title_authors_affiliations.pdf”

To the best of our knowledge, we believe that our manuscript adheres to PLOS ONE’s style requirements. 

Some of the authors’ affiliations were re-ordered to ensure this matched the formatting guidelines. 

2. “Please provide additional details regarding participant consent. In the ethics statement in the Methods and online submission information, please ensure that you have specified (1) whether consent was informed and (2) what type you obtained (for instance, written or verbal, and if verbal, how it was documented and witnessed). If your study included minors, state whether you obtained consent from parents or guardians. If the need for consent was waived by the ethics committee, please include this information. If you are reporting a retrospective study of medical records or archived samples, please ensure that you have discussed whether all data were fully anonymized before you accessed them and/or whether the IRB or ethics committee waived the requirement for informed consent. If patients provided informed written consent to have data from their medical records used in research, please include this information.”

As stated IRB Approval was obtained from the University of Cambridge

When respondents first clicked on a link to the online survey, they were presented with an information page which explained the nature and purpose of the study. By continuing past this opening page, they gave informed consent to participate in the study. 

The methods section has been updated to reflect that this was informed consent. 

3. “Thank you for stating in your Funding Statement: “Research in MRNK's laboratory is supported by a core support grant from the Wellcome Trust and Medical Research Council to the Wellcome Trust-Medical Research Council Cambridge Stem Cell Institute. This report is independent research arising from a National Institute for Health Research Clinician Scientist Award to MRNK (CS-2015-15-023). The funders had no role in study design, data collection and analysis, decision to publish, or preparation of the manuscript.” Please provide an amended statement that declares *all* the funding or sources of support (whether external or internal to your organization) received during this study, as detailed online in our guide for authors at http://journals.plos.org/plosone/s/submit- now. Please also include the statement “There was no additional external funding received for this study.” in your updated Funding Statement. Please include your amended Funding Statement within your cover letter. We will change the online submission form on your behalf.”

Please see the amended funding statement at the end of this document. There were no additional sources of funding for this study and the statement was amended to reflect that. 

4. “Thank you for providing the following Funding Statement: “BMD is supported by NIHR POLYFIX DCM and NIHR Clinical Doctoral Research Fellowship grants. BMD is a founder of MoveMed (a digital therapeutics platform which develops assessments and treatments using software). The funders had no role in study design, data collection and analysis, decision to publish, or preparation of the manuscript.” We note that one or more of the authors is affiliated with the funding organization, indicating the funder may have had some role in the design, data collection, analysis or preparation of your manuscript for publication; in other words, the funder played an indirect role through the participation of the co-authors. If the funding organization did not play a role in the study design, data collection and analysis, decision to publish, or preparation of the manuscript and only provided financial support in the form of authors' salaries and/or research materials, please review your statements relating to the author contributions, and ensure you have specifically and accurately indicated the role(s) that these authors had in your study in the Author Contributions section of the online submission form. Please make any necessary amendments directly within this section of the online submission form. Please also update your Funding Statement to include the following statement: “The funder provided support in the form of salaries for authors [insert relevant initials], but did not have any additional role in the study design, data collection and analysis, decision to publish, or preparation of the manuscript. The specific roles of these authors are articulated in the ‘author contributions’ section.” If the funding organization did have an additional role, please state and explain that role within your Funding Statement. Please also provide an updated Competing Interests Statement declaring this commercial affiliation along with any other relevant declarations relating to employment, consultancy, patents, products in development, or marketed products, etc. Within your Competing Interests Statement, please confirm that this commercial affiliation does not alter your adherence to all PLOS ONE policies on sharing data and materials by including the following statement: "This does not alter our adherence to PLOS ONE policies on sharing data and materials.” (as detailed online in our guide for authors http://journals.plos.org/plosone/s/competing-interests). If this adherence statement is not accurate and there are restrictions on sharing of data and/or materials, please state these. Please note that we cannot proceed with consideration of your article until this information has been declared.”

The funding organisation played no role in the study design, data collection and analysis, decision to publish, or preparation of the manuscript. 

BMD’s contribution to the study was through project conceptualisation, supervision, and manuscript revision. This been updated within the author contributions section of the online submission form. 

The funding statement has been amended to reflect the funder’s role. The amended statement is at the bottom of this document. 

The competing interests statement has been amended to add the statement: "This does not alter our adherence to PLOS ONE policies on sharing data and materials.”

5. “Please review your reference list to ensure that it is complete and correct. If you have cited papers that have been retracted, please include the rationale for doing so in the manuscript text, or remove these references and replace them with relevant current references. Any changes to the reference list should be mentioned in the rebuttal letter that accompanies your revised manuscript. If you need to cite a retracted article, indicate the article’s retracted status in the References list and also include a citation and full reference for the retraction notice.”

To the best of our knowledge, we believe our reference list to be complete and correct. 

Reviewers’ comments:

1. Reviewer #1: “Its very well written article with sound methodology and analysis of the results. It gives good insight into DCM. This helps inn early diagnosis of the DCM. How much of importance to be given to non conventional symptoms?”

Thank you. We believe that the results from this study suggest that non-conventional symptoms are significantly more prevalent than previously thought, even at disease presentation. However, it remains unclear exactly how useful they will be in diagnosing DCM at an earlier stage. We hope that the results of this study will prompt further evaluation and investigations of this possibility. 

2. Reviewer #2: “The study design and analysis are well done. I have little to say about this. The authors focus on the symptoms, and how they do or do not relate to early diagnosis, which is an important finding. However, the frequency of first symptoms add to the body of knowledge, the top 5 to 6 symptoms that the patient reports, should lead to more in depth assessment of the patient and that general comment or statement is not made in the paper. For the group writing this paper simply reporting on results should not be the end of the paper, a recommendation or two should come of it. I would like to see something related to, "if we see ...... symptoms, further assessment should be done to confirm diagnosis." An emphasis should be placed on, when upper limb symptoms occur, the neck should be cleared and red flags should be investigated. I would like to see a little more thought and clinical relevance emphasized in the discussion and conclusion.”

Thank you. 

As this paper was intended as a descriptive and exploratory study it would be difficult to make a strong, evidence-backed recommendation for diagnostic criteria, as the prevalence of the presenting symptoms in the general population was not investigated. 

However, it is likely that many of the most common DCM presenting symptoms (especially those of the conventional pain category) will have a low specificity for DCM given their high prevalence in the general population. 

The most sensible and intuitive approach would be to consider a diagnosis of DCM when a patient presents with a combination of a conventional pain symptom with either an upper limb sensory or lower limb motor symptom. However, this approach would require validation in a properly designed diagnostic study to determine its sensitivity and specificity for detecting DCM. 

This is a goal of a working group called the RECODE-DCM Diagnostic Criteria Incubator, and remains a work in progress. We have included a recommendation to this effect in the conclusion of the paper. 

As discussed in the paper, we also feel that further exploration of non-conventional symptoms may be of benefit to early diagnosis.

---

## [Editor Report · Decision Letter 1]

3 Feb 2023

Targeting earlier diagnosis: what symptoms come first in Degenerative Cervical Myelopathy?

PONE-D-22-10626R1

Dear Dr. Davies,

We’re pleased to inform you that your manuscript has been judged scientifically suitable for publication and will be formally accepted for publication once it meets all outstanding technical requirements.

Kind regards,

Andrea Martinuzzi

Academic Editor

PLOS ONE
---

## [Editor Report · Acceptance letter]

23 Mar 2023

PONE-D-22-10626R1 

Targeting earlier diagnosis: what symptoms come first in Degenerative Cervical Myelopathy? 

Dear Dr. Davies:

I'm pleased to inform you that your manuscript has been deemed suitable for publication in PLOS ONE. Congratulations! Your manuscript is now with our production department. 

Kind regards, 

on behalf of

Dr. Andrea Martinuzzi 

Academic Editor

PLOS ONE